# Bee Pollen: Clinical Trials and Patent Applications

**DOI:** 10.3390/nu14142858

**Published:** 2022-07-12

**Authors:** Jari S. Algethami, Aida A. Abd El-Wahed, Mohamed H. Elashal, Hanan R. Ahmed, Esraa H. Elshafiey, Eslam M. Omar, Yahya Al Naggar, Ahmed F. Algethami, Qiyang Shou, Sultan M. Alsharif, Baojun Xu, Awad A. Shehata, Zhiming Guo, Shaden A. M. Khalifa, Kai Wang, Hesham R. El-Seedi

**Affiliations:** 1Department of Chemistry, College of Science and Arts, Najran University, P.O. Box 1988, Najran 11001, Saudi Arabia; jsalgethami@nu.edu.sa; 2Department of Bee Research, Plant Protection Research Institute, Agricultural Research Centre, Giza 12627, Egypt; aidaabd.elwahed@arc.sci.eg; 3Department of Chemistry, Faculty of Science, Menoufia University, Shebin El-Kom 32512, Egypt; m_h_elashal@yahoo.com (M.H.E.); hanan.hmed557@gmail.com (H.R.A.); esraaelshafiey8@gmail.com (E.H.E.); 4Plant Protection Department, Faculty of Agriculture, Assiut University, Assiut 71526, Egypt; eslamomar@aun.edu.eg; 5Zoology Department, Faculty of Science, Tanta University, Tanta 31527, Egypt; yehia.elnagar@science.tanta.edu.eg; 6General Zoology, Institute for Biology, Martin Luther University Halle-Wittenberg, Hoher Weg 8, 06120 Halle, Germany; 7Alnahalaljwal Foundation Saudi Arabia, P.O. Box 617, Al Jumum, Makkah 21926, Saudi Arabia; ahmed@alnahalaljwal.com.sa; 8Second Clinical Medical College, Zhejiang Chinese Medical University, Hangzhou 310053, China; sqy133@126.com; 9Biology Department, Faculty of Science, Taibah University, Al Madinah P.O. Box 887, Saudi Arabia; ssharif@taibahu.edu.sa; 10Food Science and Technology Program, BNU-HKBU United International College, Zhuhai 519087, China; baojunxu@uic.edu.cn; 11Avian and Rabbit Diseases Department, Faculty of Veterinary Medicine, University of Sadat City, Menoufia 22857, Egypt; awad.shehata@pernaturam.de; 12PerNaturam GmbH, An der Trift 8, 56290 Gödenroth, Germany; 13Prophy-Institute for Applied Prophylaxis, 59159 Bönen, Germany; 14School of Food and Biological Engineering, Jiangsu University, Zhenjiang 212013, China; guozhiming@ujs.edu.cn; 15Department of Molecular Biosciences, The Wenner-Gren Institute, Stockholm University, S-106 91 Stockholm, Sweden; shaden.khalifa@regionstockholm.se; 16Institute of Apicultural Research, Chinese Academy of Agricultural Sciences, Beijing 100093, China; 17Pharmacognosy Group, Department of Pharmaceutical Biosciences, Uppsala University, Biomedical Centre, P.O. Box 591, SE 751 24 Uppsala, Sweden; 18International Joint Research Laboratory of Intelligent Agriculture and Agri-Products Processing, Jiangsu Education Department, Jiangsu University, Nanjing 210024, China; 19International Research Center for Food Nutrition and Safety, Jiangsu University, Zhenjiang 212013, China

**Keywords:** bee pollen, diseases, functional foods, cosmetics

## Abstract

Bee pollen is a natural cocktail of floral nectar, flower pollen, enzymes, and salivary secretions produced by honeybees. Bee pollen is one of the bee products most enriched in proteins, polysaccharides, polyphenols, lipids, minerals, and vitamins. It has a significant health and medicinal impact and provides protection against many diseases, including diabetes, cancer, infectious, and cardiovascular. Bee pollen is commonly promoted as a cost-effective functional food. In particular, bee pollen has been applied in clinical trials for allergies and prostate illnesses, with a few investigations on cancer and skin problems. However, it is involved in several patents and health recipes to combat chronic health problems. This review aimed to highlight the clinical trials and patents involving bee pollen for different cases and to present the role of bee pollen as a supplementary food and a potential product in cosmetic applications.

## 1. Introduction

Recently, consumer expectations of natural products have been rising due to their proven nutritional value. Food manufacturers are also responding to the health trend of “you are what you eat”, especially in the areas of functional and health-beneficial products, for both prevention and treatment. The honeybee (*Apis mellifera* L.) produces several beneficial entities, including honey, propolis, royal jelly, bee venom, bee pollen, beehive air, and beebread. These products have been used in traditional medicine for thousands of years, and there is increasing interest in their application in modern medicine [1,2,3,4,5,6,7,8,9].

Bee pollen is in line with this trend, and has great potential to contribute to research and development [10]. National legislation governs the nutritional content of bee pollen (Poland—PN-R-78893 ‘Obno a pykowe’—Polish bee pollen legislation; Switzerland—Swiss Food Manual: Pollen Bienenprodukte, BAG—Swiss Federal Office for Public Health) [11]. Bee pollen is a combination of floral nectar, flower pollen, and enzymes, as well as salivary secretions from honeybees [4]. It contains a wide range of secondary metabolites, including proteins, carbohydrates, fatty acids, vitamins, polyphenols, phytosterols, carotenoid pigments, enzymes, and co-enzymes. Pollen’s secondary metabolites have antibacterial, antioxidant, anti-atherosclerotic, anticancer, antiallergenic, anti-fungicidal, chemopreventive, hepatoprotective, and immunomodulatory effects [12].

Bee pollen has played a role in combating metabolic disorders such as diabetes, obesity, hyper-dyslipidemia, and related cardiovascular complications [4].

In addition to the nutritive value of bee pollen, it also has a physicochemical composition (water, protein, and lipid content) and techno-functional properties (protein solubility, carbohydrate solubility, and emulsifying ability) that facilitate its food application. Additionally, its capacity to absorb oil allows it to function as a flavor retainer and mouthfeel enhancer, and contribute to consistency characteristics and the creation of product structures [13].

Bee pollen products have recently been developed as granules, pills, candy bars, oral liquids, and human tonics [14]. The natural antioxidant capacity of bee pollen enhances food preservation via the prevention of lipid oxidation [15]. Bee pollen is recognized as a medicinal commodity in Germany [16]. The usage of bee pollen in yogurt increases the shelf life and improves its appearance, taste, odor, and cohesiveness [17].

Moreover, bee pollen could be used in the cosmetic field, protecting cells from abnormal melanogenesis in skin disorders, and eliminating age spots, freckles, melasma, and malignant melanoma [18].

The goal of this review is to evaluate the current status of the dietary and health impacts of bee pollen, especially on human health, with a particular emphasis on its aid in the treatment of certain ailments, as well as contemporary applications in the food sector and clinical trials. Additionally, the main recommendations for bee pollen use in cosmetic applications will be discussed.

## 2. Method

A screening of clinical and patent applications involving bee pollen was carried out. The keywords “bee pollen, clinical trials” and “patents” were used. A thorough literature search was conducted using Sci-Finder, PubMed, Google Patents, Clinicaltrials.gov (accessed on 1 January 2022), and Google Scholar [19].

## 3. Clinical Trials

### 3.1. Prostatitis Diseases

Bee pollen extracts are proven to be effective against undesirable prostatic conditions due to their anti-inflammatory and anti-androgen properties [20]. In a study, 90 patients aged 19 to 90 years, with chronic prostatitis syndrome, received pollen extracts to check the possibility of alleviating the complications (Cernilton N “pharma stroschein”, licensed by Cernitin SA, Lugano, Switzerland; Hamburg, Germany) in the form of one tablet, three times a day, for 6 months. The patients were categorized into two groups, 72 without any complications and 18 with complications, where the complications were categorized as urethral strictures, prostatic calculi, or bladder neck sclerosis. Overall, 26 (36%) participants with no complicating factors were cured of their symptoms and signs, 56 (78%) had an overall favorable clinical response, and 30 (42%) improved significantly, with an increase in flow rate, decrease in leukocyturia in post-prostate massage urine (VB3), and a decrease in C3/coeruloplasmine content in the ejaculate [21]. Similarly, 60 individuals with chronic non-bacterial prostatitis/chronic pelvic pain syndrome were randomized to receive either prostate/polit or a placebo for 6 months in a double-blind, placebo-controlled study. Three daily prostate/polit tablets, each containing 74 mg of highly defined pollen extract from Gramineae species, were given. The placebo was the same but without pollen extract. The patients who received prostate/polit tablets healed or improved, in contrast to those who received the placebo, with no adverse effects [22].

One of the most abundant flavonoids in bee pollen is quercetin and quercetin therapy lowers inflammation, as measured by prostaglandin E2 levels and increases endorphin levels [23]. One capsule of quercetin weighing 500 mg was received by patients twice daily for 4 weeks. Quercetin is suggested to play a major role in combating chronic prostatitis by lowering the oxidative stress metabolite F2-isoprostane in the patient’s expressed prostatic secretions (EPS) (Figure 1) [23]. 

A double-blind, placebo-controlled clinical trial was conducted in 47 patients with benign prostatic hyperplasia (BPH) to investigate the efficacy and safety of the 12-week intake of food supplemented with honeybee-collected pollen lump extract (HPLE). The participants were randomly assigned to three study groups: placebo or group P (0 mg HPLE per day), lower dose or group L (160 mg HPLE per day), and high dose or group H (320 mg HPLE per day). The maximum flow rate was significantly higher in group H (P0.05), but not in groups L or P. While the residual urine volume increased significantly in groups L and P (P0.05 each), it decreased in group H. There were no HPLE-related health risks or clinically significant laboratory abnormalities found [24].

Cernitin pollen extract was also administered to 79 patients suffering from BPH. The patients’ ages ranged from 62 to 89 years (mean, 68 years), and Cernitin pollen extract was taken three times a day at a dose of 126 mg (2 tablets, 63 mg each), for more than 12 weeks. Symptom scores decreased significantly, and this positive trend continued throughout the treatment period. The maximum and average flow rates of urine increased from 9.3 mL/s to 11 mL/s and 5.1 mL/s to 6 mL/s, respectively. The volume of residual urine decreased significantly, from 54.2 mL to less than 30 mL. The prostatic volume remained unchanged. However, after more than a year of treatment, 28 patients had a mean decrease in prostatic volume of 26.5 cm^3^, with no negative reactions observed. The overall clinical efficacy was 85% [25].

### 3.2. Cancer Diseases

The anticancer impact of bee pollen administration is explained by the observed underlying mechanism, such as the stimulation of apoptosis, the inhibition of cell proliferation in multiple cell lines, and the reduction of tumor growth [26,27,28]. Lotus (*Nelumbo nucifera)* bee pollen can induce apoptosis and inhibit the proliferation of human prostate cancer PC-3 cells (Figure 1) [29].

The utilization of bee pollen and honey in 46 patients >18 years old, with breast cancer and receiving antihormonal treatment, was reported in a clinical study. The patients were advised to consume a tablespoonful of pollen and honey mixture for a consecutive 14 days. In breast cancer patients receiving antihormonal therapy, honey and bee pollen ameliorated the expected menopausal symptoms [30]. The impact of non-estrogenic pollen extract PCC-100 on 300 women between 50 and 65 years old, who suffered from vasomotor symptoms during adjuvant hormonal treatment for breast cancer, was estimated in a double-blind study. PCC-100, has been indicated to treat vasomotor symptoms in postmenopausal women receiving adjuvant hormone therapy, as well as menopausal women who have never developed breast cancer [31].

### 3.3. Allergic Diseases

Grass pollen immunotherapy is effective in patients with summer hay fever, attributed to its ability in preventing the late reactions brought about by allergens [32]. Forty-four patients with severe summer hay fever participated in a randomized, double-blind, placebo-controlled, parallel-group study. The patients received injections of a depot grass pollen vaccine in a rapid up-dosing cluster, for four weeks, followed by monthly injections for two years (Figure 1). Grass pollen immunotherapy improved the quality of life in those with seasonal allergic rhinitis and decreased the symptoms of seasonal asthma and hyperresponsiveness in the lungs [33,34]. In a double-blind clinical trial, subcutaneous grass pollen immunotherapy using *Phleum pretense* was administrated to 18 patients. Compared to the placebo group, the treated group showed significantly reduced symptom and medication scores after one year of immunotherapy. Allergen-specific IgG antibodies induced by immunotherapy can disrupt the formation of allergen IgE complexes that bind to antigen-presenting cells [35]. In an in vivo study, bee pollen phenolic extract (BPPE) (200 mg/kg) from *Eucalyptus*, *Cecropia*, *Eupatorium*, and *Mimosa* together was tested on ovalbumin-sensitized mice. The extract reduced the anti-allergic activity of the cells via the reduction of IgE and IgG1 and inhibited cell migration to the pulmonary cavity [36].

### 3.4. Skin Diseases

Bee pollen has at least 200 active compounds, allowing it to be used in cosmetics. Because of its ability to strengthen and seal capillaries, as well as its sebo-balancing action, it could be employed in the formulation of cosmetics such as creams, shampoos, and conditioners (Figure 1). Furthermore, it prevents fungal development, and, hence, bee pollen is widely used in anti-dandruff shampoos [37]. The bee pollen from acorn trees (*Quercus acutissima*) has a high content of phenolic acids, contributing to the recognized anti-melanogenesis and antioxidant activity. The underlying mechanism of action is thought to be via the inhibition of tyrosinase activity, a key enzyme in melanin synthesis [38]. Similarly, in an in vitro study that compared the impact of free (FPE) and bound (BPE) phenolic extracts of rape bee pollen on melanogenesis, FPE showed a stronger ability in protecting cells from abnormal melanogenesis compared to BPE. This action was attributed to the suppression of cyclic adenosine monophosphate cAMP, downregulation of microphthalmia-associated transcription factor (MITF), and blockage of antioxidant and anti-tyrosinase TYR pathways [18].

The impact of *Dactylis glomerata* pollen on cutaneous symptoms in individuals with atopic dermatitis was assessed in a placebo-controlled, single-center, randomized, parallel-group trial with 18 participants aged 18 to 65 years; there was a change in SCO Ring Atopic Dermatitis (SCORAD) between day 1/baseline and day 3, but the study did not mention the impact of pollen as having a positive or negative effect on cutaneous symptoms [39].

## 4. Patents of Bee Pollen

### 4.1. Prostatitis Diseases

Prostatitis, or inflammation of the prostate, is a common condition that can be caused by bacterial or non-bacterial pathogenic causes [40]. In an in vivo study, pollen was found to have the ability to suppress various inflammatory pathways, including nuclear factor kappa-light-chain-enhancer of activated B cells (NF-κB), prostaglandin E2 (PGE2), and malondialdehyde (MDA) [40]. *Brassica campestris* pollen could help rats with prostate hyperplasia by affecting the expression of miRNAs including rno-miR184, which rose as the prostate improved [41]. In addition, four weeks of administration of bee pollen or date palm suspension (100 mg/kg) improved male reproductive parameters such as testis weight, testosterone, luteinizing hormone (LH) and follicle stimulating hormone (FSH), as well as spermatogenesis, motility, and viability, in STZ-induced diabetic Wistar male rats [42].

As stated in Table 1 and Figure 2, sixteen patents were filed between 1994 and 2021 on the use of bee pollen to improve prostate function, in the form of tablets, capsules, suspensions, powders, drops, and solutions. The composition includes bee pollen coupled with other natural sources (e.g., plants and bee products). The formulations can be used to treat several disorders associated with prostatic hyperplasia by reducing the prostate volume and alleviating lower urinary tract symptoms. The formulation is used for reducing the effects of prostate inflammation on male sexual function and enhancing male sexual function [43,44,45,46,47,48,49,50,51,52,53,54,55,56,57,58].

### 4.2. Diabetes Disease β

Bee pollen contains phenols and flavonoids, which inhibit carbohydrate-hydrolyzing enzymes such as amylase and glucosidase, as well as carbohydrate absorption in the small intestine, and decrease blood glucose levels significantly (Figure 2). Furthermore, postprandial blood glucose levels were lower after bee pollen intake [59]. In mice with diabetes mellitus, oral administration of pectic bee pollen polysaccharide from *Rosa rugosa Thunb* (Rosaceae) (RBPP-P) improved diabetic symptoms and protected the pancreas (Type 1). RBPP-P increased insulin secretion and functions through the stimulation of key transcription factors MafA and Pdx1 in cells. RBPP-P also increased β-cell proliferation, and upregulated the phosphorylation levels of p38, ERK, and AKT [60]. The phenolic compounds from bee pollen *Camellia sinensis* L. extract, including 3-*O*-[2′,6′-di-*O*-(trans-p-coumaroyl)-*β*-D-glucopyranosyl]kaempferol, 3-*O*-[6′-*O*-(trans-*p*-coumaroyl)-*β*-D-glucopyranosyl]kaempferol, and gallic acid (GA), had a hypoglycemic effect for patients with type 2 diabetes. Through interactions with glucose transporters, the three phenolic compounds decrease glucose absorption and transport. In addition, molecular docking showed that phenolic compounds have the ability to form hydrogen bonds with _D_-glucose and amino acids [61].

As shown in Table 2, patents reveal that bee pollen has a positive influence on sugar regulation. The combination of bee pollen and propolis had a considerable impact on diabetes patients’ blood glucose levels, with no side effects [62]. Bee pollen, Philippine flemingia root, *Radix Astragali*, and *Radix Puerariae* were incorporated into a pharmaceutical formulation to help reduce blood sugar levels [63]. A bee pollen and propolis formulation was tested on 28 hyperglycemic individuals. For 12 and 23 patients, the blood sugar level was lower after one week and two weeks, respectively [62]. In addition to other ingredients, bee pollen lipopenicillinase cold tea, rehmannia rhizome, and selenium could help to cure diabetes by protecting beta cells in the pancreas from oxidative damage, promoting sugar metabolism, and lowering blood sugar and glucose levels in urine [64].

### 4.3. Immunity-Related Disorders

The active compounds in bee pollen are vital for boosting the number and activity of humoral immune cells and phagocytes, increasing the number of red blood cells, accelerating antibody formation, and delaying the elimination of antibodies (Figure 2) [72]. Bee pollen polysaccharide CCP-1 and CPP-2 isolated from bee pollen *Crataegus pinnatifida* Bge improved the phagocytic rates and phagocytic indexes of macrophages. Moreover, CCP-2 stimulated splenocyte proliferation and NK cells [73]. Four hundred birds were fed with bee pollen at concentrations of 0, 0.5, 1, and 1.5% for five replicates in a fully randomized model. Immunoglobulin M (IgM) titers increased linearly with bee pollen dietary intake for 21 days, and similarly, thymus weight increased linearly with bee pollen dietary intake for 42 days, indicating that up to 1.5 percent bee pollen could be added to broiler feed until the age of 21 days to improve bird immunity [4,74].

According to the patents published, bee pollen in combination with other materials has a positive impact on immunity, as shown in Table 3. Corn pollen polysaccharide promotes immune organ growth in the spleen, bone, lymph nodes, and thymus gland, and moreover boosts immunocyte activity and improves the body’s ability to fight bacteria and viruses [75]. Oral liquid administration of wolfberry bee pollen is easily absorbed by the human body and has anti-fatigue and immune-boosting effects [76]. More than 20 amino acids and polyphenols, including flavone compounds, sterols, and polysaccharides, are found in *Fructus lycii* bee pollen, which are beneficial for immunological function and additionally have anti-aging and anticancer effects [77] (Table 3).

### 4.4. Chronic Diseases

Polyphenols found in bee pollen have antioxidant and antiproliferative properties, as well as the ability to regulate cell proliferation and cause apoptosis (Figure 2) [28]. The steroid fraction of bee pollen derived from *Brassica campestris* chloroform extract induced apoptosis in prostate cancer PC-3 cells, resulting in cytotoxicity [109]. It has been reported that bee pollen has a synergistic effect with the chemotherapy drug cisplatin, which is used to treat breast cancer, and that it might be used as a supplement during treatment [28]. Melissa pollen displayed a substantial impact on the treatment of breast diseases when administered once a day, as demonstrated in Table 4 [110]. The capsule/tablet also included wall-broken bee pollen and *Ganoderma lucidum* spore powder, which acts as an anti-tumor agent [111].

Heart and coronary artery diseases are examples of cardiovascular diseases. Atherosclerosis is an inflammatory and reactive process in the arteries, associated with high serum cholesterol, oxidative stress, blood clotting, and a disrupted renin–angiotensin–aldosterone system equilibrium. For 16 weeks, 54 ApoE-knockout female mice were fed diets rich in bee pollen ethanolic extract (dosage 0.1 g/kg body mass). The levels of total cholesterol (TC), asymmetric di-methylarginine (ADMA), oxidized low-density lipoprotein (ox-LDL), angiotensin-converting enzyme (ACE), and angiotensin-converting factor (ACEF) decreased significantly [4,112].

As shown in Table 4, bee pollen stimulates microcirculation and leads to lipid reduction. In patients with cardiovascular and cerebrovascular disease, a bee pollen formulation had a curative effect. *Salviae miltiorrhizae*, bee pollen, and kudzuvine root were used in a mixture that improved cardiovascular disease symptoms [113].

**Table 4 nutrients-14-02858-t004:** Different forms of bee pollen for chronic diseases.

Inventor	Country	Ingredients/Patent Form	Title/Patent Number/Patent Office	Features and Benefits	References
Ding, H.	China	Parts of raw materials by weight: 20–30 bee pollen, 20–30 bee placenta, 10–15 propolis, 20–25 royal jelly, 20–30 honey, 10–15 *Rabdosia rubescens*, 5–10 *Pseudo-ginseng*, 5–10 Ganoderma applanatum pat, 3–8 Lucid Ganoderma, 5–10 Acanthopanax, 3–8 peach kernels, 3–8 *Rheum officinale,* and 3–8 cassia twig/ND	Bee product emulsion for preventing cancer/CN 103239523 A/CN	Reduce inflammation, promote cancer resistance.	[114]
Yu, N.	China	Bee pollen, Ganoderma powder, soya lecithin, and melatonin/tablet or capsule	A series of compound health foods and its preparation method/CN 1277028 A/CN	Anti-tumor, reducing lipids.	[115]
Chen, J.	China	By wt. parts: 6–8 Urtica fissa herb, 10–16 bioflavonoid, 7–11 *Achillea millefolium*, 3–7 *Lycium chinense* and/or 2–4 bee pollen, *Lycium barbarum* fruit, 5–9 *Foeniculum vulgare* fruit, 12–16 *Salvia miltiorrhiza* root, 4–8 Dioscorea opposita and/or *Dioscorea batatas* rhizome peel, 10–14 calcium carbonate, 2–6 *Panax ginseng* ext., and 5–7 *Momordica charantia* fruit powder/ND	A kind of natural additive for foodstuffs/CN 108420070 A/CN	Prevents cancer, lowers blood lipids and blood glucose.	[116]
Xiong, B.	China	Bee pollen, pine pollen, semen, natural calculus bovis, radix euphorbiae lantu, airpotato yam rhizome, tripterygium hypoglaucum, medical stone, momordicae, cortex cinnamomi, realgar, arsenic, processed semen strychni, bulb of edible tulip, dried lacquer, raw rhizoma pinelliae, garden balsam seeds, venenum bufonis, euphorbia kansui, euphorbia pekinensis roots, roots and leaves of wikstroemia indica, moleplant seed, rhizoma anemarrhenae, muskmelon pedicel, chalcanthite, gamboge, borneol, pseudobulbus cremastrae seu pleiones, semen pharbitidis, seaweed, kelp, roasted pangolin, white muscardine silkworm, sea clam powder, dragon bone and conch shell, ND	Preparation method of external navel plaster for adjuvant therapy for cancer/CN 112755125 A/CN	It is utilized as an auxiliary treatment for cancer.	[117]
Yu, J.; Wei, Y.	China	Parts by weight: 0.3–0.5 bee pollen, 100–120 rice, 8–12 wheat grass, 8–12 celery, 4–8 melon seeds, 4–8 lemon, 4–8 honey, 6–10 *Ziziphus Jujuba*, 4–5 wheat flour, 0.5–0.7 black rice powder, 0.4–0.6 corn meal, 0.6–1 Mori Folium, 0.2–0.3 polygonum grass, 0.2–0.4 Glycyrrhiza uralensis Fisch, and water/ND	Celery health sake and its brewing method/CN 106554886 A/CN	Cancer inhibition, prevention of cardiovascular diseases, reducing blood lipids and blood sugar.	[118]
Huang, F.	China	Parts by weight: 20–70 bee pollen, 30–120 stearoyl lactate, 30–130 biological flavone, 20–140 carragheen, 30–120 pickled red pepper, 30–150 hyacinth bean, 20–80 yellow tribute green pepper, 10–60 kudzu-vine root powder, 10–65 *Radix Salviae* Miltiorrhizae, 10–75 dietary fiber/ND	Kind of environmentally friendly food additive/CN 108294287 A/CN	Anticancer, antioxidant, anti-corrosion, reducing blood lipids and hypoglycemia.	[119]
Wang, S.	China	By wt. parts: 1.40–2 bee pollen, 24–26 pork, 36–38 asparagus, 4.50–5.50 green Chinese onion, 1.40–1.60 coriander, 0.06–0.010 monosodium glutamate, 1.10–1.30 sodium chloride, 1.10–1.30 peanut oil, 1–1.50 *Auricularia auricula*, 0.30–0.50 grape seed oil, and 0.25–0.35 soy sauce/dumpling	Asparagus dumpling with healthcare function/CN 108272000 A/CN	Prevents cancer cells from dividing and growing, accelerates detoxification, lowers blood pressure, and improves cardiovascular functions.	[120]
Liu, G.	China	Parts of raw materials by weight: 80–120 Melissa pollen, 30–60 castor bean, 200–400 egg, 3–8 bee venom, 30–60 propolis, and 200–400 honey/powder	A traditional Chinese medicinal composition for treatment of breast disease and its preparation method/CN 105380982 A/CN	Good therapeutic impact in the treatment of breast diseases, particularly breast cancer, neoplasms, and hyperplasia. Low cost; it is used as a dietary treatment once a day; no side effects.	[110]
Zheng, J.	China	Parts by weight: 40–100 wall-broken bee pollen, 20–60 wall-broken *Ganoderma lucidum* spore powder, 80–150 *Lycium barbarum* fruit ext., and 5–30 western medicine such as chlorambucil, busulfan, melphalan/capsule or tablet	Pharmaceutical composition with anti-tumor activity, and its preparation/CN 105056238 A/CN	It has anti-tumor activity.	[111]
Zhang, L.	China	Parts of raw materials by weight: 45 bee pollen, 35 parts of *Salviae miltiorrhizae*, and 20 kudzuvine root/powder	Pharmaceutical composition of Salvia miltiorrhiza and bee product/CN 103800433 A/CN	Effective for cardiovascular patients.	[113]
Wu, J	China	Parts by weight through soaking in Chinese liquor: 15–30 pollen pini 20–40 honeybee room, 20–30 *Giant salamander bone*, and 10–20 *Ampelopsis grossdentata*/ND	One kind of anti-aging health wine and its preparation method/107034106 A/CN	Tumor inhibition, improves gastrointestinal functions, protect liver, and anti-aging impact.	[121]

ND: Not detected.

### 4.5. Microbial Diseases

Because bee pollen comprises flavonoids and phenolic acids, its ethanol extracts are beneficial against Gram-positive and Gram-negative bacteria such as *Staphylococcus aureus*, *Escherichia coli*, *Klebsiella pneumoniae*, and *Pseudomonas aeurgionsa*, as well as fungi such as *Candida albicans* [122,123]. The usage of bee pollen and propolis inhibits the growth and reproduction of bacteria and microbes, as shown in Table 5 and Figure 2 [124]. Wall-broken bee pollen honey wine is a stronger antibacterial agent that can be used to limit the growth and reproduction of dangerous bacteria such as *Helicobacter pylori* [125]. Bee pollen contains antibacterial and antioxidant components such as flavones and polysaccharides, and the flavone component can also suppress COX-2 activity in gingival tissues, making it an excellent anti-inflammatory agent [126]. Furthermore, bee pollen is utilized in toothpaste because of its ability to suppress germs and also reduce inflammation in the mouth, teeth, and gums, thus preventing and treating oral inflammation, and increasing oral immunity [127].

### 4.6. Applications in Food Industry

Bee pollen is used in food processing due to its nutritive, chemical, physical, and techno-functional properties (Figure 2). It has higher oil absorption capacity than water absorption capacity, low protein but high carbohydrate solubility, better emulsifying properties, and foam depressing activity [13]. Black pudding with bee pollen is a natural antioxidant source to prevent lipid oxidation [130]. Moreover, its addition to pineapple juice at 400 MPa increased the overall bioactive compounds such as phenolic and carotenoids within 15 min [131]. Due to its content of proteins, including essential amino acids, enzymes, coenzymes, large numbers of vitamins, and trace elements [132], bee pollen was used as a food supplement for older horses and prevented the reduction in hematological parameters seen in control horses; it increased the homeostasis of several lipid parameters, and improved the homeostasis of urea and plasma proteins [133]. Table 6 illustrates how its stability can make it attractive for daily consumption, and it is highly recommended for diabetes patients [134].

### 4.7. Cosmetic Applications

Rose bee pollen has a substantial impact on acne therapy when taken as tablets, oral liquid, capsules, electuary, recreational food, or a beverage. It contains proteins, minerals, vitamins, vital amino acids, and fatty acids such as linoleic and linolenic acids [147]. Melissa bee pollen acts on skin cell trophism, wrinkle reduction, as well as freckle therapy [148]. Ganoderma bee pollen extract maintains skin moisture and smoothness, improves skin tension, and maintains skin youth (Table 7 and Figure 2) [149].

## 5. Concluding Remarks and Future Perspectives

Pollen from bees has been used since prehistoric times due to its remarkable medical potential. Bee pollen has gained considerable interest because of its proven nutritional value, particularly in the fields of functional and health-beneficial manufacturing. Many ailments, including diabetes, cancer, cardiovascular diseases, prostatitis, and microbial and immune diseases, have been treated with bee pollen. The use of bee pollen in bread, fried, stewed, and canned foods, pastries, and beverages increases the nutritional value of the product, as well as its antioxidant and sterilizing properties. Scientists should apply the results from patent reports in clinical trials.

However, there are limitations in the use of bee pollen-based products due to their complexity and variability, which highlight the need for standardization before safe therapeutic usage.

## Figures and Tables

**Figure 1 nutrients-14-02858-f001:**
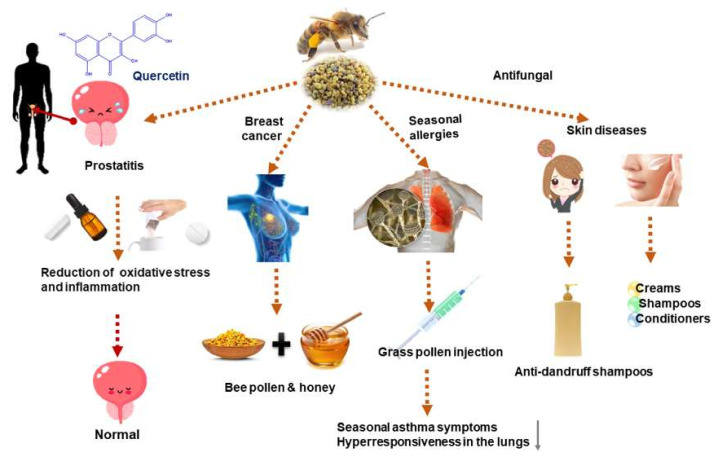
Clinical trials involving bee pollen.

**Figure 2 nutrients-14-02858-f002:**
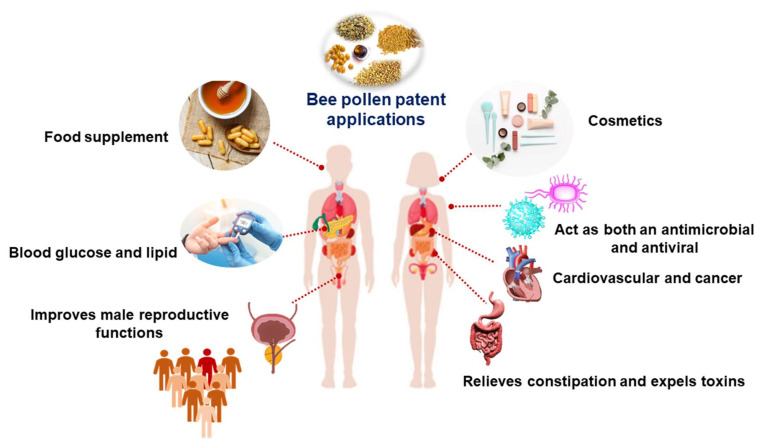
Bee pollen patent applications.

**Table 1 nutrients-14-02858-t001:** Bee pollen formulations to improve prostatic functions.

Inventor	Country	Ingredients/Patent Form	Title/Patent Number/Patent Office	Features and Benefits	References
Xu, Y.	China	Rape bee polle ND	Rape flower alkali and application thereof/CN 106,366,014 A/CN	Inhibitory effect on prostatitis (in vitro)	[43]
Chen, G.; et al.	China	Bee pollen of *Brassica*, *Poria*, *Semen Persicae*, *Ramulus Cinnamomi*, *Cortex Moutan*, and *Radix Paeoniae* Alba/Suspension, powder, drop, or solution	A Chinese medicinal prescription for treating benign prostatic hyperplasia/CN 101,590,126 A/CN	Reducing prostate volume and improving symptoms of lower urinary tract; can be used to treat some diseases related to prostatic hyperplasia, such as prostatic hyperplasia, syndromes of the lower urinary tract accompanied by prostatic hyperplasia, and other syndromes of the lower urinary tract.	[44]
Hu, A.	China	Bee pollen, pine pollen, Chinese chive seed, raspberry, medlar, *Lophatherum gracile*, honeysuckle, licorice and orange peel, chestnut flour, carrot powder, mung bean flour, Semen Coicis powder, lotus seed powder, lycopene powder, spirulina powder, wakame powder, and vitamin C powder/ND	A prostate dietetic therapy formula or replacement product/CN 103,622,006 A/CN	Reduce the incidence of prostate-related disorders.	[51]
Gou, C.	China	Bee pollen extract, Maca extract, yak penis extract, Tibetan sheep kidney extract, silkworm chrysalis extract, Lumbricus extract, trepang extract, Colla corii asini extract, bee milk extract, black wolfberry extract, an enzyme, omega-3 fatty acid, lycopene, zinc, and similar are used as raw materials/tablets	Lepidium meyenii-containing kidney-tonifying and prostate disease-treating tablet/CN 104,984,330 A/CN	Treatment for male sexual dysfunction, benign prostatic hyperplasia, prostatitis, and discomfort.	[52]
He, P.; Liu, L.; Zhang, Q.; Zhang, Y.	China	Parts by weight: 6 bee pollen, 10 pilose antler, 10 bull penis, 10 deer penis, 10 Poria cocos, 8 Chinese yam, 6 dandelion, 10 Chinese chive seed, 10 golden cypress, 6 dogwood, 6 cowherb seed, 8 pine nut, 10 oyster, 8 Cistanche, 4 chestnut, 4 ginkgo, 6 wolfberry fruit, and 25 honey/ND	Functional nutritional food for improving male sexual function and preparation method thereof/CN 113,197,301 A/CN	Minimizing the impact of prostate inflammation on sexual function. Improving male sexual function via enhancing sperm quality, increasing patients’ erection firmness, extending sexual life, and increasing female satisfaction.	[53]
Ma, H.; Wang, Y.; Kuang, H.; Lu, X.	China	80–90% rape bee pollen, 5–7% honeycomb extract, 0.15–0.30% propolis, 2–4% honeysuckle, 2–3% dandelion, 2–3% Chinese violet, and 2–3% wild chrysanthemum/capsule	Capsule for treating prostate and preparation method thereof/CN 111,228,369 A/CN	Used as treatment for prostate disease.	[54]
Qian, Z.	China	50% bee pollen, 30–70% drone pupa, and 40% queen bee	A health care product for inhibiting prostate hyperplasia and its preparation method, containing male pupa apis, larva of queen bee, and/or pollen/CN 1,432,370 A/CN	Inhibiting prostate hyperplasia.	[55]
Jiang, M.; Jiang, L.	China	Bee pollen and honey/bee pollen honeyed pill	Bee pollen honeyed pill and its preparation method/CN 1,596,710 A/CN	Good for prostate patients.	[56]
Bai, W. A.	China	35–50% bee pollen, 35–50% propolis powder, and 10–15% adjuvant material/capsule	Capsule prepared from propolis powder and bee pollen and its preparation method/CN 101,116,497 A/CN	Rehabilitation for many diseases including prostatic hyperplasia, prostatitis, and prostate diseases.	[57]
Wu, X.; Wu, Z.	China	Auxiliary material (by wt. parts): 15–25 soybeanbean, 8–12 flaxseed, 6–9 tomato, 15–25 snow pear, 10–20 tea oil, 8–12 bee pollen, and 10–20 tea leaves powder/raw materials	Preparation method for prostate health tea/CN 107,372,942 A/CN	Effective for prostate healthcare. Non-toxic, the cost is inexpensive, and is available as raw materials.	[58]
Liu, Y.	China	wt. parts: 60 bee pollen, 30 propolis, 20 royal jelly freeze-dried powder/raw materials	A Kind of bee product composition and preparation method thereof/CN 108,902,852 A/CN	Good effect in the treatment of prostatic disorders.	[45]
Li, Z.	China	wt. parts: 2–4.50 bee pollen 1.5–3.5 propolis, 2–4.5 royal jelly, and 1.5–3.8 queen bee grub, 2–4 starch, 0.2–0.8 dextrin, 0.2–0.8 powder sugar/coated tablets	Health food of purified propolis, royal jelly, bee pollen, and queen bee grub for treating prostatic diseases/CN 101,480,412 A/CN	Treat prostatitis and prostatic hypertrophy.	[46]
Mao, R.	China	50% bee pollen form Cole, 35% Pueraria root powder, and 15% auxiliary materials including microcrystalline cellulose, lactose, sodium carboxymethyl cellulose, and starch/tablet	Health tablet containing rape bee pollen and Maca powder and preparation method thereof/CN 103,622,020 A/CN	Preventing prostate from hypertrophy, controlling internal secretion sexual function, and increasing prostate function, fertility.	[47]
Li, Z.; Wu, K.; Tan, Z.	China	wt. parts: 100–150 bee pollen, 50–100 propolis, 20–80 queen bee larva, 20–80 royal jelly, 50–100 honey, and 100–150 *Parasitic loranthus* extract/ND	Method for manufacturing health composition of bee products and Taxillus chinensis/CN 101,836,736 A/CN	Increasing the prostatic, sexual, and gastrointestinal functions.	[48]
Guo, L.	China	80% bee pollen (rape pollen or corn pollen) and 20% *Cucurbita moschata*/powder	Method for manufacturing pollen product for treating chronic prostatitis, prostate dysfunction, and prostatauxe/CN 101,869,592 A/CN	Treating chronic prostatitis, functional disorders of the prostate and prostate hypertrophy.	[49]
Wang, M.; Lei, Q.	China	Lucid Ganoderma–pollen liquor consists of 100 parts ethanol water of 25~60% (weight), 3~15 parts bee pollen, 1~10 part Ganoderma, 0.1~0.3 part Fructus Lycii, 0.1~0.2 part organic acid esters, 0.05~0.1 part sweetening agent/alcoholic medical health drink	A medicated liquor containing Ganoderma and bee pollen/CN 1,096,444 A/CN	Effective for conditions such as prostate hyperplasia and constipation.	[50]

ND: Not detected.

**Table 2 nutrients-14-02858-t002:** Different forms of bee pollen for diabetic diseases.

Inventor	Country	Ingredients (Parts by Weight)/Patent Form	Title/Patent Number/Patent Office	Features and Benefits	References
Cheng, Y.; et al.	China	By weight of: 50–70 *Trichosanthes kirilowii*, 3–8 Lilii Bulbus, 5–8 mulberry, 3–5 glutinous rice flour, 5–7 Sargassum powder, 10–15 *Puerariae radix* lyophilized powder, 1–2 strawberry essence, 3–6 bee, 1:5–7:3–5 propolis dry powder, broken bee pollen, and royal jelly powder/powder	A hypotensive and hypoglycemic Fructus trichosanthis powder and processing method thereof/CN 105,963,566 A/CN	Assists treatment of diabetes. Long-term consumption can improve patient’s immunity and promote health.	[65]
Zhang, K.	China	By weight part: 45 bee pollen, 35 Trichosanthes root, and 20 root of kudzu vine/powder	Medicine composition containing *Radix trichosanthis* and bee product/CN 103,893,254 A/CN	Decreasing blood glucose and lipid levels.	[66]
Pang, Y.	China	By weight part: 45 bee pollen, 35 *Philippine flemingia* root, 33 *Radix Astragali*, and 20 *Radix Puerariae/* powder	One kKind of pharmaceutical composition with bee product and Flemingia/CN 104,398,566 A/CN	Effective in lowering lipid and glucose levels.	[63]
Ma, N.; Chai, H.	China	Bee pollen, papain, amylase, lipase, saccharifying enzyme, Sargassum extract, Panama extract, *Citrus reticulata* polyphenol, tetrahydropiperine, *Gynostemma pentaphyllum*, *Cyclocarya paliurus*, Radix Trichosanthis, Rhizoma Coptidis*, Litchi chinensis*, *Trifolium pratense*, *Agrimonia pilosa*, *Portulaca oleracea*, *Ophiopogon japonicus*, *Momordica charantia*, anion powder, IR ceramic powder, and heat-sensitive agent/ND	External composition for treating diabetes mellitus and preparation method thereof/CN 106,237,315 A/CN	Regulating blood sugar.	[67]
Wu, G.; Fan, J.; Wang, F. F.	China	Weight part: 3–7 bee pollen, 22–27 ginger, 32–40 *Polygonati Rhizoma*, 12–17 medlar, 5–13 *Puerariae Radix* powder, 0.8–2.5 magnesium stearate/tablet	One kind of fine white ginger compound health buccal tablet and its preparation method/CN 106,539,022 A/CN	Regulating blood glucose level, improving immunity, and anti-aging effect.	[68]
Liu, X.; Liu, H.; Liu, H.; et al.	China	By wt. part: 20–40 bee pollen and 20–60 propolis/powder	A blood sugar-reducing propolis pollen and preparation method thereof/CN 113,080,408 A/CN	Significant impact in reducing blood glucose. For diabetic patients. No side effects.	[62]
Gao, C.	China	Raw materials by weight: 5–15 g bee pollen, 5–15 g Chinese yam, 4–15 g Poria cocos, 5–15 g of dried orange peel, 4–15 g oyster, 4–15 g raspberry, 4–15 g radish seed, 3–12 g *Cordyceps militaris*, 5–15 g dandelion, 5–15 g *Perilla frutescens*, 5–15 g gorgon fruit, 5–15 g perilla seed, 5–15 g coix seed, 5–15 g purslane, 5–15 g *Polygonatum odoratum*, 2–10 g *Angelica dahurica*, 5–15 g lily, 1–10 g donkey-hide gelatin, 5–15 g lotus seed, 2–10 gjujube, 2–10 g dark plum, one cactus, 0.5–5 g pseudo-ginseng, 1–5 g safflower carthamus, 1–5 g *Astragalus mongholicus*, 2–10 g *Oldenlandia diffusa*, 2–10 g barbed skullcap herb, 10–20 gRhizoma polygonati, 2–12 g kudzu root, 10–20 g fleece-flower root, 4–15 gbalsam pear, 4–15 g medlar, 2–10 g *Potentilla discolor*, 4–15 g *dwarf lilyturf* tuber, and 2–15 g of mulberry leaf/powder	Medicine for treating diabetic complications and preparation method thereof/CN 112,451,629 A/ CN	Treatment for complications of diabetes. It has no side effect, with good impact on lowering appetite.	[69]
Yao, Z.	China	Milk tea has raw material of prebiotic content by following parts by weight: 4–8 bee pollen, 10–20 prebiotic compositions, 10–15 skimmed milk power, 20–30 passionfruit, 80–100 tea base, 20–30 ginkgo leaf, 4–8 juice powder, 4–8 yeast, 4–8 dragonfruit, 4–8 *Gynura procumbens* (Lour.) Merr extract, 8–12 *Pleurotus eryngii*, 1–5 lower of *kudzuvine*, 1–5 Radix bardanae, 1–5 bluish dogbane, 1–5 cassia seed/ND	With prebiotic composition milk tea and its preparation method/CN 107,114,507 A/CN	Effective impact on regulating blood glucose level, decreasing lipid level, and enhancing digestion.	[70]
Bin, Y.	China	Weight ratio: 10–15 bee pollen,15–20 dried rhizome of Rehmannia rich in selenium, 5–10 bee glue powder, 8–12 *Radix paeoniae* rubrathe, 6–10 *Radix Astragali*, and 3–5 root of Chinese trichosanthes/ND	One kind of se-rich Rehmannia glutinosa bee pollen lipid-lowering herbal tea/CN 104,336,233 A/CN	Decreasing blood lipid and blood glucose. Simple preparation.	[64]
Wu, J.; Zhang, Y.	China	4–6 bee pollen share, 8–10 peanuts, 6–8 Hippophae fructus powder, 5–8 Hominis placenta powder, 4–6 Astragali radix powder share, 3–5 Maydis stigma 5–8 share, wolfberry/form/ND	With Lowering Blood Pressure Multifunctional Health Product and Preparation Method Thereof./CN 106,551,357 A/CN	Blood pressure lowering. No toxicity, no side effects, short course, quick, effective, and low-cost as healthcare product.	[71]

ND: Not detected.

**Table 3 nutrients-14-02858-t003:** Different forms of bee pollen for immunity-related disorders.

Inventor	Country	Ingredients/Patent Form	Patent Title/Patent Number/Patent Office	Features and Benefits	References
Di, D.; Pei, D.	China	0.01–0.2 g/mL wall-broken bee pollen, 0.01–0.2 g/mL *Lycium barbarum*, 0.01–0.1 g/mL honey, 80 0.0005–0.003 g/mL tween, 0.0004–0.005 g/mL xanthan gum, glyceryl monostearate 0.0005–0.005, carrageenan 0.0004–0.005, sodium alginate 0.0004–0.005, citric acid 0.002–0.008 g/mL, and water/oral liquid	Oral liquid containing bee pollen and Lycium barbarum and manufacture method thereof/CN 102,742 894 A/CN	Improves immunity and fatigue resistance, and nourish the skin.	[76]
Di, D.; Pei, D.; Liu, J.	China	Parts by wt.%: 40–50 wall-broken *Fructus lycii* bee pollen, 15–20 *Radix codonopsis* extract, 10–15 lyophilized powder of spirulina, 5–10 filler, 3–5 lubricating agent, and 1–2 essence/ND	Anti-tumor health-care food containing fructus lycii bee pollen and preparation method thereof/CN 109315739 A/CN	Improve immunity throughout human body.	[77]
Zhang, J.	China	By wt.%: 10–95 bee pollen and 5–90 *Gynostemma pentaphyllum*/tablet	Method for manufacturing health care food containing gynostemma pentaphyllum and bee pollen/CN 102302107 A/CN	Increase human immunity. Decrease blood pressure, blood lipid, and sugar. Prevent tumors by inhibiting and killing cancer cells.	[78]
Shao, S.	China	Parts by wt.: 150–180 bee pollen, 10–30 plant extract, 10–20 pupa worm intestinal extract, 10–20 modified dietary fiber, 7–15 oligosaccharide, 2–6 *Lactobacillus plantarum* powder, 2–6 hydroxyl isomaltulose, and 2–6 mango powder/tablets, electuary, capsules	One kind of bee pollen for improving immunity and its preparation method/CN 105,831,759 A/CN	Improves immunity.	[79]
Liang, J.	China	bee Pollen, taizishen, longan tree parasitic, mushroom, and amino acids/ND	A Chinese medicine composition to improve immunity and preparation method thereof/CN 104257722 A/CN	Improving immune function. Quick impact and simple in preparation and execution.	[80]
Miao, J.	China	Parts by wt.: 20–30 bee pollen, 30–50 honey, 20–30 honey lyophilized powder, 15–25 soybean lecithin powder, 10–20 spinach, 10–20 oligofructose, and 5–10 low-fat milk/capsule	Capable of improving immunity of honey pollen capsule/CN 106,666,595 A/CN	Improves immunity, high nutritional value, low cost, and simple to prepare	[81]
Miao, J.	China	Parts by weight: 20–30 bee pollen, 30–45 honey, 20–30 honey, 20–25 Radix Astragali, 15–20 mushroom, 10–20 matrimony vine, 10–20 parts of FOS, 5–10 American ginseng, 25 ferrous lactate/capsule	A kind of honey pollen capsule for fatigue/CN 106,962,862 A/CN	Improves immunity, fatigue resistance.	[82]
Liu, J.; Sui, Y.	China	Parts by wt.: 1–2 bee pollen, 10–20 kiwi flower, 5–10 blueberry, 50–100 honey, 2–5 honeysuckle, 2–4 Cortex eucommiae male flower, and 5–10 camellia/powder	Flower tea product and preparation method thereof/CN 112,868,847 A/CN	Enhance immunity, prevent cardiovascular diseases, antibacterial, anti-inflammatory, and hepatoprotective properties. Easy to prepare and low cost.	[83]
Ding, H.	China	Parts by wt.: 10–40 bee pollen, 20–60 honey, 20–60 royal jelly, 1–10 bee glue, 20–30 bee placenta, 2–9 ginseng, 3–12 pseudo-ginseng, 5–18 dogwood fruit, 3–12 *Rhizoma alismatis*, and 5–20 Chinese wolfberry/ND	Bee-product-containing health product/CN103,238,771 A/CN	Improves immunity, expel toxins, beautify the face, anti-aging, and beneficial for the brain, kidneys, tendons, and for bone strengthening.	[84]
Liu, J.	China	Parts by wt.: 0.01–0.015 bee pollen, 70–90 wheat flour, 5–15 wheat malt powder, 0.001–0.005 lotus root polysaccharide, 0.006–0.01 selenium-enriched yeast, 1.8–2.2 peanut sprout powder, 0.1–0.5 modifier, 0.005–0.008 corn peptide, and 0.015–0.02 white tea powder/ND	Corn peptide selenium-rich polypeptide nutrition flour/CN 104,000,080 A/CN	Improves immunocompetence.	[85]
Bai, W.	China	By wt.%: 35–50 bee pollen (purity > 98 wt.%), 35–50 propolis powder (purity > 90 wt.%), 2–4 VC, 10–15 and auxiliary additive/capsules	Health food with vitamin C/propolis compound capsules/CN 102,132,810 A/CN	Improves immunity; regulate blood pressure; fatigue and insomnia resistance. High nutrition, high active ingredient content, low dose, low cost, and no side effects.	[86]
Liu, Y.	China	By wt. %: 68–72 bee pollen, 8–12 honey, 18–22 *Ampelopsis grossedentata*, and water for balance.	Method for manufacturing healthcare tea containing bee pollen, honey, and Ampelopsis grossedentata/CN 102,224,867 A/CN	Improves immunity, decrease blood pressure, help in body weight loss, and inhibit laryngopharyngitis.	[87]
Zhao, J.	China	By weight percent: 60–95 bee pollen, 5–30 *Ginkgo biloba* extract, and 0–10 starch/tablets or capsules	Method for preparing healthcare food from *Ginkgo biloba* extract and bee pollen/CN 101,669,640 A/CN	Improves immunity, prevent cardiovascular diseases, decrease lipid level, anti-aging, and beautifying skin.	[88]
Chen, Wi; Chen, X.	China	By wt.: 10–40 bee pollen, 20–60 honey, 20–60 royal jelly, and 1–10 propolis/ND	Formula of bee products for improving immunity/CN 102,125,196 A/CN	Improves immunity.	[89]
Yin, W.; Yin, Z.	China	Weight of parts: 10–60 bee pollen, 2–20 propolis, 2–20 royal jelly, 2–15 bee embryos, 2–15 been pupae, and 2–10 *Panax quinquefolium/*ND	Method for manufacturing honey product preparation for improving immunity/CN 102,150,770 A/CN	Improves immunity, decrease blood lipid, fatigue resistance, and anti-aging.	[90]
Zheng, Z.	China	Parts of raw materials by weight: 1–5 bee pollen, 10–15 *Salvia miltiorrhiza*, 12–18 bioflavonoid, 5–10 soybean functional polysaccharide, and 2–6 *Fructus lycii*/ND	Natural food additive/CN 111,134,318 A/CN	Improves immunity and inhibit cancer. Decrease blood sugar and blood lipids in human body. Used as food additive due its antioxidant and bacteriostasis effects.	[91]
Shao, S.	China	Parts by weight: 50–70 bee pollen of maize extract, 35–50 corn oligopeptide powder, 10–20 modified dietary fiber, 10–20 fruit vegetable powder, 5–20 pectin decomposer, 3–12 chrysanthemum powder, 2–10 sophora flower powder, 2–10 isomalt/ND	One kind of bee pollen for preventing and treating liver disease and enhancing the immune system and its production method/CN 104686883 A/CN	Improves immunity, and protect liver.	[75]
Ren, X.	China	Parts by weight: 8–12 bee pollen, 4–6 medlar, 4–6 yam, 5–7 chrysanthemum, 4–6 Gordon euryale seed, 1–5 pearl barley, 2–4 red bean, 2–6 prepared Rehmannia root, 4–8 dogwood, 4–8 tuckahoe, 1–3 tree peony bark, and 0.5–1.5 oriental waterplantain rhizome/granule powder	Nutrient instant powder/CN 111671082 A/CN	Improve immunity, enhancing sleep, and anti-aging. Used as supplement.	[92]
Zhong, H.	China	Wt. in parts: 3–5 lotus bee pollen, 3–5 linden bee honey, 1.0–1.5 protease, 1.0–1.5 wine yeast, and 88–92 purified water/ND	Method for manufacturing healthcare wine from honey and pollen/CN 102911834 A/CN	Strengthen immunity, support to the liver and kidneys, lower blood pressure, and moisturize the skin.	[93]
Liang, Z.	China	By weight percent: 0.5–3 bee pollen, 5–18 *Pseudo-ginseng*, 5–12 *Ginkgo biloba* extract, 12–22 glucose syrup, 18–55 granulated sugar, 4–15 powdered milk, and 10–18 petal/powder	Fragrant health candy containing Panax notoginseng for resisting fatigue, anoxia, and aging, as well as reducing blood fat and sugar/CN 102058005 A/CN	Improves immunity, combats exhaustion and oxygen depletion, anti-aging, anticancer; decrease blood fat and sugar. It has nutritive value, good fragrance, and stable quality.	[94]
Liu, J.; Yin, Y.	China	Wt. parts: 5–25 bee pollen, 25–55 royal jelly, 15–35 honey, and 15–35 wolfberry powder/ND	Composition for improving sexual function of men/CN 105433328 A/CN	Enhance immunity and improve male sexual function.	[95]
Wang, L.; Sui, L.	China	By wt. parts: 3–9 bee pollen, 1–5 *Chrysanthemum morifolium*, 1–5 *Shizhu Panax* ginseng, 3–9 *Acanthopanax sessiliflorum*, 3–9 *Poria cocos*, 3–9 *Sesamum indicum*, 1–5 *Semen Persicae*, 1–5 *Tremella fuciformis*, 1–5 *Auricularia auricula*, 1–5 *Hericium erinaceus*, 2–8 *Lentinus edodes*, 1–5 grape seed, 0.02–0.06 lycopene, and 0.5–1.5 royal jelly/ND	Health food with antioxidation function and its manufacture method/CN 101518336 A/CN	Enhance immunity.	[96]
Li, W.; Liu, S.	China	Fresh milk and bee pollen as main materials, adding fractions of sucrose, nourishing sugar (cerealose or glucose), and maltodextrin/milk powder	A health promotion milk powder containing bee pollen\CN 1108477 A\CN	Increased immunity and improved human physiology.	[97]
Inventor not announced	China	By wt. parts: 10–90 *Rabdosia serra* and 10–90 bee pollen/tablets, granules, and capsules	Healthcare food made from Linearstripe isodon herb and bee pollen and processing method thereof/CN 102302106 A/CN	Detoxification, delaying aging, protecting the liver, and increasing human immunity.	[98]
Inventor not announced	China	By wt. parts: 10–40 *Berchemia lineata* and 60–90 bee pollen/tablets, capsules, and granules	Method for manufacturing healthcare food containing Berchemia lineata and bee pollen/CN 102302105 A/CN	Increase human immunity. Good for patients with tuberculosis, diabetes, gastric ulcers, testitis, spermatorrhea, rheumatoid arthritis, lumbar genu ache, traumatic injury, and urticaria.	[99]
Inventor not announced	China	By wt. parts: 60–95 bee pollen and 5–40 lycopene/tablets, granules, and capsules	Method for manufacturing healthcare food containing lycopene and bee pollen/CN 102302104 A/CN	Increase immunity, treat heart and cerebrovascular diseases, and good for patients with tumors, hepatitis, anemia, and diabetes.	[100]
Li, G.; Li, H. F.	China	By wt.%: 65–85 bee pollen (purity >= 80%) and 15–35 active casein peptides (purity >= 80%)/ND	Process for manufacturing health food containing peptides and bee pollen/CN 101889667 A/CN	Delay aging, improves immunity, and resist fatigue.	[101]
An, X.	China	500–800 g of bee pollen and 1000–1500 g of mature honey/ND	For prevention and treatment of malignant tumor for traditional chinese medicine compound preparation and preparation method thereof/CN 106344621 A/CN	Improves gastrointestinal functions, good supplement for humans, beneficial for sleep. Strengthening chemoradiotherapy effects, improving immune function for patients with malignant tumors.	[102]
Long, L.; Wei, Z.	China	0.1–10% bee pollen, 0.1–20% bamboo leaf extract, and 70–99.8% carrier/feed additive	Feed additive containing bee pollen and bamboo leaf extract for improving immunologic function of livestock and poultry/CN 109170172 A/CN	Significantly improves immunity of livestock and poultry. Promotes absorption and utilization of fat-soluble vitamins and other microelements. Novel feed additive, which is safe, efficient, and environmentally friendly.	[103]
Zhang, G.	China	Bee pollen, Vaccinium, and *Rosa davurica*/ND	Production of healthcare food comprising Vaccinium, Rosa davurica, and bee pollen/CN 101449824 A/CN	Enhance heart function, improve eyesight, kill bacteria, improve immunity, reduce blood fat, and combat tumors.	[104]
Zhang, Q.	China	Bee pollen, rice flour, medlar, *Zizania aquatica*, apple juice, broccoli, sweet potato powder, white sugar, coix seed flour, walnut kernel, dried orange peel, honey, and olive oil/rice cake	Nutritional rice cake/CN 109170526 A/CN	Eliminating dampness and cold, enhancing immunity.	[105]
Bai, Z.	China	Tangerine bee pollen, tangerine honey, tangerine royal jelly, fritillaria, white peony root, and*Radix Angelicae* Sinensis/ND	Traditional Chinese medicine with tangerine honey/CN 108450866 A/CN	Protecting liver and kidneys, remove sputum from lungs, and strengthen immunity.	[106]
Gou, C.	China	Bee pollen, black wolfberry, maca, sheep placenta extract, royal jelly, Rhizoma polygonati, gelatin, semen coicis, walnuts, collagen, γ- linolenic, vitamin C, vitamin E/tablet	Lycium ruthenicum tablet as health product\CN 104126744 A/CN	Increase immunity, antioxidant, anti-aging, delay menopause, protect the liver, and has anti-tumor effect.	[107]
Lin, J	China	Raw material drug by weight: 1–2 rose bee pollen, 1–2 *Fructuctus schisandrae* bee pollen, 1–2 honeycomb extract, 1–2 *END*, 1–2 Eclipta alba, 1–2 *Radix paeoniae alba,* and 1–2 *Prunella vulgaris*	A bee pollen health product and method for preparation/CN 104083464 A/CN	Significantly controls endocrine secretion, improves immunity, relieves menopausal symptoms, enhances sleep, reduces migraines, reduces blood pressure, and decreases weight.	[108]

ND: Not detected.

**Table 5 nutrients-14-02858-t005:** Different forms of bee pollen that acts against antimicrobial diseases.

Inventor	Country	Ingredients/Patent Form	Title/Patent Number/Patent Office	Features and Benefits	References
Cao, J.	China	wt. parts: 50–80 bee pollen, 25–50 dry lotus leaves, 35–65 parched hawthorn fruit, 30–50 dried plum-leaf crab, 5–15 dried orange peel, 1–15 medicated leaven, 1–10 *Rhizoma Atractylodis Macrocephalae*, 1–10 *Radix Codonopsis*, 10–20 corn silk, 2–10 Chinese wax gourd peel, 20–30 wolfberry fruit, and 1000–3000 liquor/ND	*Nelumbinis folium* alcohol/CN 106701480 A/CN	Antibacterial and antiviral activity.	[128]
Li, B.; Chen, H.	China	Parts of raw materials by weight: 7–9 bee pollen, 46–52 mushroom dregs, 28–32 malt sprouts, 16–20 soluble glass, 9–12 propolis, 85–95 bred maggots, 0.7–0.8 trypsin, 125–135 rice bran, 0.9–1.3 alpha amylase, 5–7 urea phosphate, and a sufficient amount of water/ND	Antiviral pollution-free plant nutrient solution/CN 105399564 A/CN	Antiviral activity.	[124]
Mi, Y.	China	Parts of raw materials by weight: 5–15 g rose bee pollen, 5–15 g rape bee pollen, 5–15 g buckwheat bee pollen, 5–15 g corn bee pollen, 5–15 g lotus bee pollen, 5–15 g hundred flower bee pollen, 60–80 g honey, and 500–600 g Fenjiu honey wine/ND	Wall-broken bee pollen honey wine and preparation method thereof/CN 111454811 A/CN	Enhancing metabolism, inhibition of *Helicobacter pylori* growth.	[125]
Cao, G.	China	0.1–2% bee pollen liquid, 40–60% sorbitol, 20–40% silicon dioxide, 5–15% deionized water, 1.5–4% sodium lauryl sulfate, 0.8–1.5% sodium carboxymethyl cellulose, 0.2–0.4% titanium dioxide, 0.1–0.2% saccharin sodium, and 0.1–0.3% benzyl alcohol/toothpaste	Bee pollen healthcare toothpaste/CN 104706549 A/CN	Antibacterial and anti-inflammatory impact on teeth and gums.	[127]
Song, H.; Wang, H.	China	2–5% enzymolysis bee pollen, 2–5% glycerol, 0.5–1% vitamin C, 0.3–0.5% hydrolyzed fish skin collagen, 0.5–1% menthol, 1.5–2.5% surfactant, 0.1–0.5% sodium carboxymethylcellulose, 0.01–0.1% preservative, 0–1% sweetening agent, and a balance of water/mouthwash	Mouthwash for gingivitis prevention and treatment/CN 113633604 A/CN	Gingivitis prevention.	[126]
Qiu, H.	China	4–9 bee pollen, 40–50 Semen Maydiss, 25–30 Testa Tritici, 20–25 bean cake, 14–18 Fructus Cucurbitae moschatae, 6–15 *Oleum Glycines*, 7–13 yeast rich in zinc, 9–16 fishbone powder, 1–3 phosphoric acid hydrogen calcium, 0.8–1.5 threonine, 5–8 Chinese medicine extraction liquid, 1–4 salt, 3–5 Lactobacillus, 6–12 wheat meal stone, 3–5 garlicin, and 2–6 saccharin sodium/feed additive	A kind of pregnant pig feed/CN 106472843 A/CN	Antibacterial and antiviral properties. Great impact on tocolysis in midwifery.	[129]

ND: Not detected.

**Table 6 nutrients-14-02858-t006:** Patent forms of bee pollen in food applications.

Inventor	Country	Components/Form	Title/Patent Number/Patent Office	Features and Benefits	References
Jiang, Y.; Xie, Z.; Zhang, H.G.; et al.	China	By weight parts: 20–40 bee pollen, 30–35 *Hordeum vulgare* powder, 25–40 lactose, and 1–3 green tea extract/ND	Healthcare Chinese medicinal composition for increasing dietary fiber, facilitating feces excretion, and removing toxins/CN 105901693 A/CN	Increased dietary fiber, thus aids constipation relief and toxin expulsion.	[135]
Mi, Y.	China	Weight of parts: 1–5 bee pollen, 10–15 fruit, 12–18 bioflavonoid, 5–10 soybean functional polysaccharide, 2–6 medlar, 12–15 red bean, 8–12 hawthorn, 4–8 carotene, 15–20 nisin, 8–15 Cordyceps sinensis powder, 20–25 red-silk thread, and 10–15 essence/ND	Antibacterial food additive and its preparation process/CN 113424959 A/CN	Antiseptic ability, good mouthfeel and taste as food additive.	[136]
Yin, Y.; Zhang, Y.	China	Raw materials in parts by mass: 12–22 bee pollen, 20–40 Sargassum horneri sodium alginate, 20–30 defatted soybean protein, 15–40 dextrin, 18–25 pickled red pepper, 18–20 grape seed extract, 15–18 Huanggongfjiao, 8–55 malt or wheat germ, 6–10 Radix puerariae powder, and 5–8 Salvia miltiorrhiza/ND	A type of environmentally friendly food additive and production method/CN 111955728 A/CN	Food additives, good quality of the product.	[137]
Taihe, B.; Jizhou, C. S.	China	wt. parts: 92–95 Lini pollen, 5–8 Lini anthophorids/ND	Production method of α-linolenic acid food additive/CN 104643099 A/CN	Alpha-linolenic acid food additive with low cost.	[138]
Liu, G.; Bai, W.; Li, N.; et al.	China	*Fructus jujubae* and camellia bee pollen/ND	Camellia bee pollen mixed jujube functional food and preparation method thereof/CN 105054037 A/CN	Health promotion and nutritive value.	[139]
Fregonese, A.	France	By weight parts: 25–50 bee pollen, 50 clay, 10–25 hypromellose, 0.1–1 titanium dioxide, and copper and chlorophyllin complex/capsule	Compositions based on clay and bee pollen, method for preparing same, and nutritional and therapeutic uses thereof/WO 2012080333 A1/FR	Food supplement.	[140]
Beu, A. I.	Romania	5–70% beeswax, 5–95% pollen, and at least one of the following: 0.1–40% propolis tincture, 10–40% honey, 0.1–20% apilarnil, up to 95% maiden wax, 0.1–10% royal jelly, up to 20% bee venom, 15% flavoring, 30% sweetener, 60% dry fruit, 80% medicinal plant powder, and 80% seeds/tablets, pills, bon-bons, dragees, drops	Method for preparing bee product-based food supplements and confectionery products/RO 126992 B1/RO	Food supplements and confectionery Products.	[134]
Chakhunashvili, K. G.; Chakhunashvili, D. K.; et al.	United States	wt. by percent: 33–40% bee pollen, 20–33% water, and 33–40% alcohol such as ethanol/powder	Bioactive food/US 20150093494 A1/US	Food additive for bread and other bakery products. It has nutritional value due to its content of amino acids, vitamins, and minerals.	[141]
Gao, Z.	China	Parts by weight: 2–4 bee pollen, 6–11 lotus leaves, 3–6 *Codonopsis pilosula*, 8–12 purslane parts, 9–11 ethyl laurate, 1.2–2 amyl butyrate, 3–5 licorice extract, 7–9 *Lycium chinense*, 3–8 emulsifier, 6–10 Chinese yam, 7–8 Chinese wolfberry fruit, 8–10 bioflavonoids, 0.2–4 Stigmata maydis, and 7–9 Chinese wolfberry fruit/ND	A food additive/CN 104970367 A/CN	Food additive, good taste and nutritional value.	[142]
Wei, X.	China	1–5 bee pollen, 10–15 salvia, 12–18 bioflavonoids, 5–10 functional soybean polysaccharides, and 2–6 wolfberry/ND	Natural food additive/CN 104431687 A/CN	Food additive for fried food, stewing food, pastries, canned food, beverage-making process. It can enhance the body’s immunity, lower blood lipids, blood sugar, and combat cancer.	[143]
Cheng, N.; Gao, H.; Cao, W.	China	Bee pollen and plain flour/ND	Method for manufacturing biscuits containing bee pollen/CN 102763701 A/CN	Used for preparing biscuits with high nutritive value, rich in flavonoid substances, improves immunity and cancer resistance.	[144]
Yuan, K.; Lin, X.; Xu, D.; et al.	China	wt. by percent: 9–20% bee pollen, 20–50% okra extract, 20–40% orange peel extract, 9–20% tea polyphenol, 9–20% spiral seaweed, and 0.0003–0.0005% selenious yeast/ND	Natural food additive containing Abelmoschus esculentus extract and wenchow orange peel extract/CN 102160646 A/CN	Food additive, good antioxidant and sterilization effects, high nutritive and healthcare value, convenient use, and easy storage and transport.	[145]
Celso de Amorim, C.; da Silva, R.; Girliane; Tania Maria; S.-S.; et al.	Portugal	Coconut monofloral bee pollen and coconut oil constituent/soft gelatinous capsules	Preparation process and formulation of soft gelatinous capsule containing coconut monofloral bee pollen and coconut oil for use as functional food/BR 102015,031440 A2/BR	Innovative formulation of a natural product of excellent quality and acceptance for dietary supplementation.	[146]

ND: Not detected.

**Table 7 nutrients-14-02858-t007:** Cosmetic applications of bee pollen.

Inventor	Country	Ingredients/Patent Form	Title/Patent Number/Patent Office	Features and Benefits	References
Wen, Y.; Hu, G. Wu, Z.; et al.	China	Camellia bee pollen, *Apricot blossom*, pea, and black rice/ND	Traditional Chinese medicine composition, traditional Chinese medicine fermented product containing same, preparation method, and applications thereof/CN 112245538 A/CN	Whitening and antioxidant cosmetics.	[150]
Wang, W.; Wang, H.	China	Corn pollen, queen bee larva freeze-dried powder, earthworm active protein, Phyllanthus emblica extg. soln., citrus hystrix extg. soln., fungus ext. soln., and pure water/ND	The preparation method of an antioxidant and anti-aging composition/CN 112205622 A/CN	Has antioxidant and anti-aging properties.	[151]
Yang, L.; Xu, G. Liu, A.; et al.	China	Moisturizing *tinctorius* bee pollen or moisturizing tinctorius inflorescence ethanol ext. or moisturizing tinctorius seed oil ext./ND	*Carthamus tinctorius* extract, cosmetics, and application by taking *Carthamus tinctorius* as active ingredient/CN 105193893 A/CN	Anti-wrinkle, whitening, sunscreen, and other skin cosmetic applications. Moisturizing and antioxidant activity.	[152]
Xu, Z.; Yin, Y.; Dong, X.; et al.	China	wt. parts: 3 rape bee pollen ext., 2–3 cetearyl glucoside, 1–2 PEG-100 stearate, 2–6 iso-Pr myristate, 2–6 caprylic/capric triglyceride, 0.5–4 di-Me siloxane, 1–5 shea butter, 2–6 cetearyl alc., 0.5–2 acrylamide/acrylate copolymer, 0.1–0.3 Me p-hydroxybenzoate, 0.1–0.3 Pr p-hydroxybenzoate, 2–8 glycerol, 0.05–0.2 EDTA-2Na, 0.05–0.2 allantoin, 0–2 essence, 0.05–0.2 diazolidinyl urea, and water in balance/cream	Rape bee pollen extract, its preparation method, and application in anti-aging cosmetic products/CN 102188350 A/CN	Acts as anti-aging facial cream. No toxicity or other side effects.	[153]
Markham, K. R.	New Zealand	A flavonoid or flavonoid deriv., i.e., baicalein, apigenin, luteolin, isoscutellarein, galangin, kaempferol, quercetin, pinocembrin, cinnamic acid, 3,4,5-trihydroxy cinnamic acid, 4-hydroxy cinnamic acid, 3,4-dihydroxy cinnamic acid, rutin, taxifolin, naringenin, or their derivs. From plants, pollen, bee propolis (propolis), esp. from *Gingko biloba*, plants of the Pinaceae family, and/or plants of the Theaceae family/cream	UV-screening composition including flavonoids or flavonoid derivatives/NZ 264108 A/NZ	Sunscreen, moisturizer, skin toner, cosmetic foundation, lipstick or cream, nail lacquer, hair shampoo, hair conditioner, hair colorant, fabric coating or protectant, laundry fabric conditioner, laundry fabric cleanser, paint stain or coating, fade-resistant ink, or dye.	[154]
Yu, N.; Yu, Q.	China	0.1–10 pine pollen, 0.1–10 pollen, 5–95 soy milk liq., 0.1–10 konjac flour, 0.1–10 *Poria cocos* powder, 1.6–20 glucomannan powder, and 3–20% xylitol powder/ND	Method for preparing composite soy milk containing micron pine pollen and konjac flour for slimming and lowering blood sugar level/CN 1785020 A/CN	Nourishes the skin.	[155]
Wang, Z.	China	Bee pollen, an edible nanometer powder of Ganoderma spore, and banana peel/nanometer granule	A nanometer powder of Ganoderma spore/CN 1640314 A/CN	Has a beneficial skincare effect.	[156]
Sun, J.; Cui, B.; Mao, X.; et al.	China	Bee pollen enzymic hydrolysate and nutrient soln., mask paper/ND	Preparation method of bee pollen facial mask/CN 111494294 A/CN	Used in cosmetics as a facial mask.	[157]
Liang, B.	China	6–7 selenium-rich Saposhnikovia divaricata, 2–3 selenium-rich Centipeda min., 3–4 *Xanthium sibiricum*, 0.5–1 *Zingiber officinale*, 3–4 *Gastrodia elata*, 2–3 grape seed ext. parts, 110–119 bee pollen, and 200–280 glucose syrup/ND	Selenium-rich cream candy for treating nasal concha swelling and preparation method thereof/CN 107568398 A/CN	Act as skin beautifying component.	[158]
Shenqing, F. Z.	China	Tannin compounds from buckwheat pollen or bee pollen including 1,2,3,4,6-penta-*O*-galloyl-b-D-glucose and 1,2,3,4,6-penta-*O*-galloyl-2-*O*-mdigalloyl-*D*-glucose/ND	Preparation of tannin compounds from buckwheat pollen and application thereof/CN 107245067 A/CN	Has a good whitening effect.	[159]
Cheng, G.; Cheng, J.	China	*Ganoderma lucidum* bee pollen vitamin, *Ganoderma lucidum* rape pollen *Scutellaria baicalensis*, anoderma lucidum sesame pollen natural fragrant grass anti-aging cream, *Ganoderma lucidum* pine pollen, *Arnebia euchroma* ext. anti-aging cream	*Ganoderma lucidum* pollen skincare anti-aging cream as cosmetic/CN 106937923 A/CN	Skincare, has good anti-radiation and anti-aging effects.	[160]
Liang, B.	China	wt. parts: 110–119 bee pollen 6–7 *Radix Stemonae*, 2–3 *Platycodon grandiflorum*, 3–4 *Xanthium sibiricum*, 0.5–1 *Zingiber officinale*, 3–4 *Gastrodia elata*, 2–3 grape seed, and 200–280 glucose syrup/ND	Toffee for treating enlargement of nasal cavity and preparation method thereof/CN 106720841 A/CN	Skincare.	[161]
Xie, J.	China	wt.: 1–5 wall-broken rose bee pollen, 1–10 aloe, 1–10 collagen, 1–5 pearl powder, 1–5 peach blossom, 1–10 Radix angelicae, 1–5 royal jelly, 1–5 lecithin, 1–5 earthworm, 0.1–1 vitamin E, 100–200 honey, and 1–8 water-soluble azone/cream	A cosmetic wrinkle-removing cream/CN 106562917 A/CN	Makes the skin delicate, increases flexibility, significantly shallows wrinkles, and makes the skin full, moisturized, and flexible.	[162]
Ruan, D.	China	Filtrate of the fermented product of two-splitting yeast, bee pollen, silk protein, elastin, protease, eggshell membrane ext., egg white ext., propolis ext., xanthan gum, milk protein, astaxanthin, hydrolyzed silk, orange oil, pearl ext. herbal ext., and deionized water/ND	Freeze-dried essence facial mask and manufacturing process thereof/CN 111773113 A/CN	Suitable for any skin type, and used for whitening, dark spots, anti-aging, moisturizing, hydrating, sterilizing, and astringent effects.	[163]
Mi, Y.	China	5–15 g rape bee pollen, 5–15 g buckwheat bee pollen, 5–15 g corn bee pollen, 5–15 g rose bee pollen, 5–15 g lotus bee pollen, 5–15 g lily bee pollen, 60–80 g honey, and 500–600 g Fenjiu wine/ND	Wall-broken bee pollen honey wine and preparation method thereof/CN 111454811 A/CN	Moistened and beautified skin tissue.	[125]

ND: Not detected.

## Data Availability

No applicable.

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
