# Peer review of "Bee Pollen: Clinical Trials and Patent Applications"

_nutrients, 2022, doi:10.3390/nu14142858_

Round 1

Reviewer 1 Report

Review notes

Article title: Bee pollen: Clinical trials and patents applications.

This manuscript reviewed the status of the dietary and health impacts of bee pollen especially on human health with particular emphasis on their aid in the treatment of certain ailments, as well as contemporary applications in the food sector and clinical trials. The manuscript is well organized and written. It seems to be a significant contribution to the field of functional and nutraceutical foods, particularly bee pollen-based products, and it could be published after minor revisions.

Specific comments are made as follows:

Most of biological activities and healthy effects of bee pollen are attributed to the presence of bioactive compounds (phenolics, flavonoids, terpenes, carotenoids, among others). In this context, please provide a table/figure/scheme of the most representative phytochemicals reported in bee pollen, followed by a brief description of them. It should be included after “Method” and before “Clinical trials”.

Table 1 to Table 7 should have an added column specifying the country of patent license.

  • Line 39: Suggested change of ¨of the honeybees¨ to ¨ from honeybees¨.
  • Line 40. Add ¨a¨ between ¨has¨ and ¨significant¨.
  • Line 43: remove ¨ ' ¨ from ¨prostates' ¨.
  • Line 45: change ¨is aiming¨ to ¨is aimed¨.
  • Line 46: change ¨food supplementary ¨ to ¨supplementary food¨.
  • Line 51: Change ¨ consumers' expectations ¨ to ¨ Consumer expectations¨
  • Line 54: Change ¨ and in line with this trend, ¨ to ¨is in line with this trend, and¨
  • Line 54: remove ¨a¨ from ¨has a great potential¨.
  • Line 56: remove ¨as in¨ from ¨ bee pollen as in¨.
  • Line 60:  change ¨included¨ to ¨including¨.
  • Line 60: change ¨fatty acid¨ to ¨fatty acids¨.
  • Line 61:  Add ¨,¨ between ¨ vitamins polyphenols¨. Please add a comma after vitamins
  • Line 66: Add ¨a¨ between ¨ has physicochemical ¨.
  • Line 70:  Change ¨ consistency characteristic¨ to ¨ consistency characteristics¨.
  • Line 70: Change ¨ products structures¨ to ¨ product structures¨.
  • Line 72: Change ¨ Bee pollen natural antioxidant capacity¨ to ¨ The natural antioxidant capacity of bee pollen¨.
  • Line 72: change ¨ the food¨ to ¨food¨.
  • Line 73. Please add a tab before [7].
  • Line 76: Add ¨the¨ between ¨ in cosmetic¨.
  • Line 81:  Change ¨ main recommendations ¨ to ¨the main recommendations ¨.
  • Line 83:  change ¨ nano-composites¨ to ¨ nano-composite¨.
  • Line 83:   Change¨ that can promote bee pollen industrial progress ¨ to ¨that can promote progress in the bee pollen industry¨.
  • Line 86:  Change ¨ the clinical and patents applications ¨ to ¨ clinical and patent applications¨.
  • Line 87: Change ¨ keyword¨ to ¨keywords.
  • Line 88-89:  There are extra spaces that need to be removed.
  • Line 93-94: Change ¨ Bee pollen extracts proved its actions against undesirable prostatic conditions owing to their anti-inflammatory and anti-androgen properties¨ to ¨ Bee pollen extracts proven to be effective against undesirable prostatic conditions due to their anti-inflammatory and anti-androgen properties¨.
  • Line 111: Suggested change of ¨ One of the abundant flavonoids in bee pollen is quercetin¨ to ¨Quercetin is a flavonoid found abundantly in bee pollen¨.
  • Line 122:  Change ¨for consecutive¨ to ¨ for a consecutive¨.
  • Line 125: Change ¨ 65 years old suffered¨ to ¨ 65 years old who suffered¨.
  • Line 132: Change “utilized” to ¨utilized¨ (the aforementioned is the word used throughout the document).  Please change
  • Line 144: Change ¨ was change¨ to ¨ was a change¨.
  • Line 145-146: Change ¨impact of pollen as it has a positive¨ to ¨impact of pollen as having a positive¨.
  • The Figure 1 (before line 148) is shifted to the right and the a few words are cut-off.
  • Figure 1: Change ¨Reduction of the oxidative stress and inflammation¨ to ¨ Reduction of oxidative stress and inflammation¨
  • Line 152: change ¨in vivo study, pollen has the ability¨ to ¨ In an in vivo study pollen was found to have the ability¨
  • Line 163: Change ¨ drops or solutions, the¨ to ¨ drops, and solutions. The¨
  • Line 166:  Suggested change of ¨recipes¨ to ¨formulation¨.
  • Line 166: add ¨are¨ before ¨used¨.
  • Table 1:  fourth column (features and benefits), second row, change ¨ improving symptom¨ to ¨improving symptoms¨.
  • Table 1:  fourth column (features and benefits), fourth row, change ¨ yperplasia¨ to ¨hyperplasia¨.
  • Table 1:  fourth column (features and benefits), ninth row, Change ¨Rehabilitation for many diseases in including¨ to ¨ Rehabilitation for many diseases including¨
  • Table 1:  fourth column (features and benefits), tenth row, Change ¨No toxic¨ to ¨ Non-toxic¨
  • Table 1:  fourth column (features and benefits), last row, Change   ¨Effective for disease cures such as and prostate hyperplasia constipation ¨ to ¨ Effective for disease cures such as prostate hyperplasia constipation¨.
  • Line 182: Add a tab after (GA)
  • Line 182-183: Add ¨a¨ between ¨ have hypoglycemic¨.
  • Line 184: Add ¨.¨ between ¨transport In¨.
  • Line 185: Add ¨a¨ between ¨ phenolics compounds¨.
  • Line 185: remove “s” from phenolics
  • Line 191: Remove ¨A” from ¨ A bee¨.
  • Line 192:  change ¨was¨ to ¨were¨.
  • Line 196: Remove ¨the¨ from¨ levels in the¨.
  • Table 2:  fourth column (features and benefits), first row, Change ¨assist¨ to ¨assists¨.
  • Table 2:  fourth column (features and benefits), seventh row, remove extra period.
  • Line 205:  Change ¨stimulate¨ to ¨stimulates¨.
  • Line 210: Change ¨ the bird immunity¨ to ¨bird immunity¨.
  • Table 3:  fourth column (features and benefits), third row, Change ¨ Increasing the human immunity. ¨ to ¨ Increasing human immunity. ¨
  • Table 3: Check column 4 for sentences that need periods (there are many).
  • Table 3:  fourth column (features and benefits), third row, Change ¨ Decreasing the blood pressure, blood lipid, and sugar. ¨ to ¨ Decreasing blood pressure, blood lipid, and sugar. ¨
  • Table 3:  fourth column (features and benefits), third row, Change ¨ Preventing tumor by¨ to ¨ Preventing tumors by¨.
  • Table 3:  fourth column (features and benefits), fifth row, Change ¨Improving the immune function. Quick in impact and simple in preparation and execute¨ to ¨Improving immune function. Quick in impact and simple in preparation and execution. ¨
  • Table 3:  fourth column (features and benefits), eighth row, Change ¨ Easy to prepare and low coast ¨ to ¨ Easy to prepare and low cost. ¨
  • Table 3:  third column (features and benefits), eighth row, Capitalize the ¨f¨ in ¨ flower¨.
  • Table 3:  fourth column (features and benefits), fifteenth row, Change ¨ Increasing the immunity. ¨ to ¨ Increasing immunity. ¨
  • Table 3:  fourth column (features and benefits), nineteenth row, Change ¨ Strengthen immunity, support the liver and kidneys, lower blood pressure, and moisturize the skin ¨ to¨ ¨ Strengthened immunity, support to the liver and kidneys, lower blood pressure, and moisturizes the skin ¨
  • Table 3:  fourth column (features and benefits), twentieth row, Change ¨ Improve immunity to ¨ Improved immunity¨.
  • Table 3:  fourth column (features and benefits), row 23-24, Change ¨Increasing immunity and improve human physiology. ¨ to ¨ Increased immunity and improved human physiology. ¨
  • Table 3:  fourth column (features and benefits), row 25,26, and 27, Change ¨ Increasing the human immunity. ¨ to ¨ Increasing human immunity. ¨
  • Table 3:  fourth column (features and benefits), row 31, Change ¨Significantly improve immunity ¨ to ¨Significantly improves immunity ¨.
  • Table 3:  fourth column (features and benefits), last row , Change ¨Significantly Controlling endo-crine secretion, im-proving immunity, relieving menopau-sal symptoms, en-hancing sleep, re-ducing migraines, reducing blood pressure and de-creasing weight. ¨ and ¨ ¨Significantly Controlls endocrine secretion, improvs immunity, relieves menopausal symptoms, enhances sleep, reduces migraines, reduces blood pressure and decreases weight. ¨
  • Line 230:  Change ¨powder acts¨ to ¨powder which acts¨.
  • Line 232: Change ¨Heart diseases and coronary artery diseases¨ to ¨Heart and coronary artery diseases¨.
  • Line 240:  Change¨ the fourth table¨ to ¨in table 4¨.
  • Table 4:  fourth column (features and benefits), third row, Change ¨Prevents cancer, lowers blood lipids and blood glucose¨.
  • Table 4: Check column 4 for sentences that need periods (there are many).
  • Table 4: check column 4, eighth row, change ¨ as diet treatment¨ to ¨as a dietary treatment¨
  • Table 4: check column 4, eleventh row, change ¨improve¨ to ¨improves¨.
  • Line 247: change ¨ such¨ to ¨such as¨.
  • Line 250-251: Wall-broken bee pollen honey wine, revisa esta, se me hace como que no debe ir asi pero no es mi rama tampoco, entonces no se con certeza.
  • Line 259: remove the ¨s¨ in ¨works¨.
  • Table 5: check column 4, frth row, add an ¨s¨ to ¨gum¨.
  • Table 5: check column 4, last row, change ¨tocolysis midwifery impact, greatly¨ to ¨ Tocoloysis midwifery have a great impact. ¨ though I would still suggest the addition of why or on what that impact is.
  • Line 261:  change ¨Bee pollen used in food ¨ to ¨Bee pollen is used in food¨.
  • Line 262-263: Change ¨ It is ability to absorb oil more than water aborption capacity¨ to ¨ It has a higher oil absorption capacity than a water absorption capacity¨.
  • Line 264: change ¨as a¨ to ¨is a¨.
  • Line 265: change ¨ It is addition on ¨ to ¨its addition in¨.
  • Line 266:  Add ¨has an¨ before ¨increase¨.
  • Line 266:  After carotenoids, Replace ¨and¨ with ¨as well as¨
  • Line 267: Remove ¨the¨ before ¨essential¨.
  • Line 270:  Add ¨it¨ before ¨increased¨.
  • Line 271-272:  Change ¨ Table 6 illustrated how the good stability make it attractive for consuming by children and advised for diabetics¨ to ¨ Table 6 illustrates how its stability can make it attractive for children consumption and advised diabetics¨.
  • Table 6: Check column 4, first row, change¨ Increasing dietary fiber, relieving constipation and expelling toxin ¨ to ¨Increased dietary fiber, constipation relief and toxin expulsion¨.
  •  Table 6: Check column 4, last row, change ¨dietarysupplementation¨ to ¨dietary supplementation¨.
  • Table 7: Check column 4, third row, change ¨application ¨ to applications¨.
  • Table 7: Check column 4, fifth row, change ¨ Act as antiaging facial cream No toxicity or other side effect ¨ to ¨ Acts as an antiaging facial cream. No toxicity or other side effects.
  • Table 7: Check column 4, row 7, change ¨ Nourish the skin¨ to ¨Nourishes the skin¨.
  • Table 7: Check column 4, row 8, change ¨Has skin caring effect¨ to ¨Has skin caring effects¨.
  • Table 7: Check column 4, row 9, change ¨Used in cosmetics as facial mask¨ to ¨Used in cosmetics as a facial mask¨.
  • Table 7: Check column 4, row 10, change ¨Act as skin beautifying¨ to ¨ Act as skin beautifying component¨.
  • Table 7: Check column 4, row 11, change ¨Has a good whitening effect¨ to ¨ Has a good whitening effect¨.
  • Figure 2: Change title to ¨Bee pollen patent applications¨. Change ¨Act as both antimicrobial and antiviral¨ to ¨Acts as both an antimicrobial and antiviral¨. Change¨ Improvement male reproductive functions¨ to ¨Improves male reproductive functions¨. Change¨ Relieving constipation and expelling toxin¨ to ¨Relieves constipation and expels toxins¨.
  • Line 282: Change ¨Patents applications of bee pollen¨ to ¨Patent applications of bee pollen¨.
  • Line 286: Change ¨owned¨ to ¨ due¨.
  • Line 287: Remove ¨today¨. Change ¨the proven¨ to ¨its proven¨.
  • Line 291: Change¨ and canned¨ to ¨canned¨.
  • Line 293: Remove ¨s¨ from ¨ patents¨.

Please could you expand the perspective section and add the main limitations of patent process of bee pollen-based products.

Author Response

Response letter  enclosed

Reviewer 2 Report

The subject chosen to be reviewed is of great importance in the academic area, encompassing the area of health and food.

The abstract is a good summary of the central idea of the manuscript, inviting the reader to continue reading. Additionaly, the objective of the work is well described and is well explored. The figures illustrate the text content very well and the tables summarize the studies evaluated by manuscript.

However, the method of searching for patents and clinical cases seems to have limited the study to its country of origin. Also, it is extremely important to explore how bees collect pollen and how pollen is extracted by beekeepers and finally transformed into a product/drug.

A robust and adequate review of the literature is necessary when it comes to a review article. My big concern is that several topics were written based on just one publication.

The article was divided into two major parts: Clinical trials and patents applications. To my surprise, the article minimally explored trials clinicals. So I suggest that authors improve the clinicals part or rewrite the article only based on patents applications.

Other important points are highlighted below:

1 – Keywords: The words bee pollen, clinical trials and patents are already cited in the title of the study. I suggest replacing them.

2 - The introduction is confusing and needs improvement. Many different subjects are presented in a single paragraph, making the reading tiresome and monotonous.

3 - In general, the studies used in this article mentioned the type of pollen used? What plant did it come from?

4 - Quercetin appears to be largely responsible for the improvement in EPS cases. Have the studies cited amounts of quercetin found in pollen given to patients?

5 - The topic "Cancer diseases" is confusing and underexplored.

6 - On the topic "Allergy diseases" is there only one published study?

7 - Grass pollen injections? How are these injections prepared? How are they applied? How much pollen is used?

8 - Skin diseases: Is there only one published study?

Author Response

Response letter  enclosed

Reviewer 3 Report

The article covers a very interesting topic and summarizes the potential of bee pollen on human health. I recommend reviewing the English. I suggest pointing out in some cases that it is the combination of bee pollen with other molecules that affects a certain disease and not exclusively the pollen.

·         Pages 2-24 lines 51-54, 59-61, 66-69, 131-132, 137-140, 174-177, 177-178, 179-183, 183-184, 202-204, 204-207, 232-235, 235-236, 240-241, 241-242, 276-277, 286-287 the bibliographical reference is missing, please add it.

·         Page 15 line 230 'Ganoderma lucidum' should be written in italics

·         Page 15 line 237 ‘TC’ please write in full and the abbreviation TC in brackets.

·         Page 18 in the table Helicobacter pylori instead of helicobacter pylori

·         Since the tables are very long, I recommend putting column headings on each page where they are located.

Author Response

The article covers a very interesting topic and summarizes the potential of bee pollen on human health. I recommend reviewing the English. I suggest pointing out in some cases that it is the combination of bee pollen with other molecules that affects a certain disease and not exclusively the pollen.

Response: The authors read the manuscript thoroughly and amended it accordingly, followed by thorough English editing

  • Pages 2-24 lines 51-54, 59-61, 66-69, 131-132, 137-140, 174-177, 177-178, 179-183, 183-184, 202-204, 204-207, 232-235, 235-236, 240-241, 241-242, 276-277, 286-287 the bibliographical reference is missing, please add it.

Response:  Adjusted

  • Page 15 line 230 'Ganoderma lucidum' should be written in italics

Response:  Adjusted

  • Page 15 line 237 ‘TC’ please write in full and the abbreviation TC in brackets.

Response:  Adjusted

  • Page 18 in the table Helicobacter pylori instead of helicobacter pylori

Response:  Adjusted

  • Since the tables are very long, I recommend putting column headings on each page where they are located.

Response:  Adjusted

Round 2

Reviewer 2 Report

The article has improved considerably. However, to be ready for publication, I suggest that the English is preferably proofread by a native speaker.